# 3D Environment Is Required In Vitro to Demonstrate Altered Bone Metabolism Characteristic for Type 2 Diabetics

**DOI:** 10.3390/ijms22062925

**Published:** 2021-03-13

**Authors:** Victor Häussling, Romina H. Aspera-Werz, Helen Rinderknecht, Fabian Springer, Christian Arnscheidt, Maximilian M. Menger, Tina Histing, Andreas K. Nussler, Sabrina Ehnert

**Affiliations:** 1Siegfried Weller Research Institute, BG Trauma Center Tübingen, Department of Trauma and Reconstructive Surgery, University of Tübingen, Schnarrenbergstr. 95, D-72076 Tübingen, Germany; victor.haeussling@student.uni-tuebingen.de (V.H.); rominaaspera@hotmail.com (R.H.A.-W.); helen.rinderknecht@student.uni-tuebingen.de (H.R.); carnscheidt@bgu-tuebingen.de (C.A.); mmenger@bgu-tuebingen.de (M.M.M.); thisting@bgu-tuebingen.de (T.H.); sabrina.ehnert@med.uni-tuebingen.de (S.E.); 2Department of Diagnostic and Interventional Radiology, University of Tübingen, Hoppe-Seyler-Str. 3, D-72076 Tübingen, Germany; fabian.springer@med.uni-tuebingen.de; 3Radiology Department, BG Trauma Center Tübingen, Schnarrenbergstr. 95, D-72076 Tübingen, Germany

**Keywords:** diabetes mellitus, bone metabolism, cryogel, scaffold, 3D coculture, osteoblast, osteoclast

## Abstract

A large British study, with almost 3000 patients, identified diabetes as main risk factor for delayed and nonunion fracture healing, the treatment of which causes large costs for the health system. In the past years, much progress has been made to treat common complications in diabetics. However, there is still a lack of advanced strategies to treat diabetic bone diseases. To develop such therapeutic strategies, mechanisms leading to massive bone alterations in diabetics have to be well understood. We herein describe an in vitro model displaying bone metabolism frequently observed in diabetics. The model is based on osteoblastic SaOS-2 cells, which in direct coculture, stimulate THP-1 cells to form osteoclasts. While in conventional 2D cocultures formation of mineralized matrix is decreased under pre-/diabetic conditions, formation of mineralized matrix is increased in 3D cocultures. Furthermore, we demonstrate a matrix stability of the 3D carrier that is decreased under pre-/diabetic conditions, resembling the in vivo situation in type 2 diabetics. In summary, our results show that a 3D environment is required in this in vitro model to mimic alterations in bone metabolism characteristic for pre-/diabetes. The ability to measure both osteoblast and osteoclast function, and their effect on mineralization and stability of the 3D carrier offers the possibility to use this model also for other purposes, e.g., drug screenings.

## 1. Introduction

Diabetes mellitus is the metabolic disorder with the highest prevalence and incidence worldwide. In 2019, the international diabetes federation (IDF) reported a prevalence of 463 million diabetics worldwide, of which approx. 95% suffer from type 2 diabetes. Estimations from the IDF indicate that this number will increase to 700 million by the year 2045. Diabetes mellitus is associated with multiple complications such as micro- and macro-vascular pathologies, nephropathy, neuropathy and alterations in bone homeostasis. The latter is often referred to as diabetic bone disease, associated with an increased fracture risk and subsequently impaired fracture healing rich in complications [1,2]. Indeed, a large British study with more than 2800 patients identified diabetes as one of the main risk factors for delayed and nonunion fracture healing [3]. Bone mineral density (BMD) is generally decreased in type 1 diabetics. Despite the increased fracture risk, most authors describe an increased BMD in type 2 diabetics [4,5], indicating towards poor quality of the bone matrix. Diabetic foot syndrome and Charcot-Osteoarthropathy are the most commonly described representatives of diabetic bone diseases, affecting approx. every 5th diabetic within the first 5 years of the disease [6]. In 20% of the cases, this results in an amputation of the affected limb [7], which represents the major cause for nontraumatic amputations in developed countries. A large proportion of the long-term therapeutic costs for diabetics accounts for the treatment of these bone diseases [8]. There is evidence that an early adjustment of the therapy cannot only reduce complications of the eyes, the kidneys, or the nervous system [9], but may also have positive effects on the progression of diabetic bone diseases. For obtaining such information, advanced model systems are required [10].

In 2012, an in vitro model mimicking the onset of a type 2 diabetes by stimulating primary human osteoblast with high concentrations of insulin and/or glucose was described. In this model, matrix mineralization was inhibited in the presence of hyperinsulinemia [11]. As this does not represent the situation in type 2 diabetics [4,5], it was assumed that the used target cells were too mature. In another in vitro model, based on immortalized human bone marrow-derived MSCs (SCP-1 cells), pre-/diabetic conditions were simulated by replacing the fetal calf serum in the medium by serum of corresponding patients. Sera from both prediabetics (obese) and diabetics strongly affected osteogenic differentiation of SCP-1 cells, also resulting in reduced matrix mineralization [12]. This raised the question whether inhibition of osteoclast function may account for the increased BMD frequently observed in type 2 diabetics [4,5]. However, this cannot be displayed in the described models, which utilized only osteogenic cells.

Therefore, aim of this study was to establish an in vitro model displaying changes in bone metabolism characteristic for type 2 diabetics, which involves both bone formation by osteogenic cells and bone resorption by osteoclastic cells. For potential use in substance screenings later on, the model should be constantly available in considerable amounts. Therefore, the model should be based on cell lines. In earlier studies, the myeloid cell line THP-1 was used to generate osteoclastic cells in an indirect coculture approach using the culture supernatant of primary human osteoblasts [13]. These cells proved to be compatible with SaOS-2 cells (osteosarcoma derived cell line), whose osteogenic features match well with mature primary human osteoblasts [14]. To better mimic the in vivo situation, direct coculture of these two cell types will not only be tested in conventional 2D culture but also in 3D culture, as the 3D conformation is thought to support the maturation and function of the bone cells applied to the carrier [15]. However, the choice of the carrier might be crucial, as it may affect the cells’ attachment to and infiltration into the carrier, as well as the cells’ differentiation [16]. Regarding the carrier, a large variety of materials have been proven to be compatible for bone tissue engineering. Poly-2-hydroxyethyl methacrylate (pHEMA)-based cryogels with different protein additives have been described to be suitable for culturing bone cells in 3D [17,18]. With a mean pore diameter of approx. 100 µm, a porosity over 50% in a hydrated state, and a stiffness of approx. 65 kPa, the pHEMA-based cryogels are within a range recommended for bone tissue engineering [17]. Although, the mineral density of the pHEMA-based cryogels (50 mg/cm^3^) is relatively low when compared to bone, the external crystallization method provides a rough surface that favors osteogenic differentiation as well as an easily accessible calcium deposit for bone resorbing cells. This cannot only be detected in quantitative CT scans, but also affects the stiffness of the populated scaffolds [17,18]. Therefore, these cryogels will be used as base for the 3D coculture. In the proposed 2D and 3D in vitro coculture model, pre-/diabetic conditions will be simulated as described [11]. Both models will then be compared regarding osteogenic and osteoclastic function, as well as formation and degradation of mineralized matrix and hence the stability of the carrier (3D).

## 2. Results

### 2.1. Coculture and 3D Environment Stabilize the Cell Culture

Osteogenic differentiation of SaOS-2 cells was usually exhausted after 10–14 days in culture, when viability significantly dropped and matrix mineralization reached a maximum [19]. Osteoclastic differentiation of THP-1 cells reached its limits already after 7 days in culture [13]. However, when differentiating SaOS-2 cells and THP-1 cells in direct 2D coculture, cells remained viable and functional for at least 14 days. The direct 2D coculture of SaOS-2 cells and THP-1 cells showed constant mitochondrial activity over the 14 days of culture. In contrast, mitochondrial activity increased, when cells were differentiated in 3D coculture (Figure 1A). As mitochondrial activity cannot differentiate between the two cell types in the coculture, cell-specific DNA content was determined. In 2D coculture DNA from both SaOS-2 cells and THP-1 cells significantly decreased, starting at day 7 of culture. In 3D coculture, the decrease in THP-1 cells’ DNA was less pronounced than in 2D coculture, and seemed to recover until day 14. SaOS-2 cells’ DNA remained stable over the entire differentiation period in 3D coculture (Figure 1B).

### 2.2. Change of Osteoblast and Osteoclast Function in 2D and 3D Coculture

As expected in 2D coculture ALP activity (early osteoblast function) rapidly decreased. This effect was delayed in the 3D coculture (Figure 2A). TRAP activity (osteoclast function) increased over the entire differentiation period both in 2D and 3D cocultures, with a stronger increase observed in 2D coculture (Figure 2B). CAII activity (osteoclast function) increased over the entire differentiation period in 2D coculture. The increase in CAII activity was a bit more pronounced in 3D coculture, and reached its plateau between 7 and 10 days of culture (Figure 2C). Cathepsin K (CTSK) production remained stable in 2D coculture and increased in 3D coculture. In 3D coculture CTSK showed peak levels on day 7 of culture (Figure 2D). After 7 days of culture life-dead-staining revealed various multinucleated cells in the direct coculture (Figure 2E).

### 2.3. Osteoclast Activity First Detectable on Day 4 of Culture

Direct coculture was chosen, that SaOS-2 cells provide soluble receptor activator of nuclear factor kappa-Β ligand (sRANKL) required for osteoclastic differentiation of THP-1 cells. While in 2D coculture sRANKL levels strongly increased from day 7 of culture on, this effect was observed 3 days earlier in 3D coculture (Figure 3A). Its antagonist osteoprotegerin (OPG) strongly increased only in 2D coculture but remained relatively stable in 3D coculture (Figure 3B). While the sRANKL:OPG ratio significantly decreased over time in 2D culture, it remained stable in 3D culture (Figure 3C). As marker for collagen production procollagen type I N-terminal propeptide (PINP) was detected. PINP strongly increased over the entire differentiation period in 2D coculture, reaching its highest level on day 14 of culture. In 3D coculture, the increase in PINP peaked around day 7 of culture (Figure 3D). As sign for active osteoclasts, collagen degradation was determined by detection of its splice product collagen-type I N-telopeptide (NTX). NTX levels were below the detection limit on day 1 of culture, but increased from day 4 of differentiation on, in both 2D and 3D coculture (Figure 3E). Osteocalcin levels (late osteogenic marker), which are basally high in SaOS-2 cells remained stable in both 2D and 3D coculture (Figure 3F).

### 2.4. Insulin and Glucose Affect Cell Viability and Cell Composition in the Cocultures

2D and 3D cocultures were differentiated in the presence or absence of high insulin levels and high glucose levels to simulate the development of a type 2 diabetes [11]. In 2D coculture, mitochondrial activity was increased by both high insulin (HI) and high glucose (HG). In 3D coculture, however, mitochondrial activity was increased only by high insulin (Figure 4A,B). In 2D coculture, this effect came along with an increased amount of THP-1 cells, while in 3D coculture pre-/diabetic conditions increased the amount of SaOS-2 cells (Figure 4C,D).

### 2.5. Pre-/diabetic Conditions Affect Function of Osteoblastic SaOS-2 Cells

ALP activity, which normally declines with increasing matrix mineralization, was significantly increased under hyperglycemic conditions after 4 days in 2D coculture. Inversely, ALP activity was decreased under hyperglycemic conditions in 3D coculture (Figure 5A,B). While in 2D coculture PINP formation was not affected by either high insulin or high glucose, in 3D coculture PINP levels were elevated under hyperglycemia when compared to hyperinsulinemia (Figure 5C,D). Osteocalcin levels remained stable in 2D coculture, but decreased under hyperglycemic and hyperinsulinemic conditions in 3D coculture (Figure 5E,F). Levels of sRANKL, required for osteoclast formation, were decreased under pre-/diabetic conditions in 2D coculture. Simultaneously, its antagonist OPG was strongly increased by insulin and glucose. In 3D coculture, however, sRANKL levels even increased under hyperglycemic conditions and the effect on OPG was less pronounced. In 3D coculture, OPG levels were mainly affected by hyperinsulinemia. The resulting sRANKL:OPG ratio was decreased under pre-/diabetic conditions, both in 2D and 3D coculture (Figure 5G–L).

### 2.6. Pre-/diabetic Conditions Affect Function of THP-1-Derived Osteoclastic Cells

As characteristics for osteoclast function, CAII and TRAP activity were determined. In 2D coculture CAII activity decreased under pre-/diabetic conditions, with a stronger effect in the presence of high glucose levels. In contrast, in 3D coculture CAII activity showed an increasing trend under pre-/diabetic conditions (Figure 6A,B). The effect of pre-/diabetic conditions on TRAP activity was less pronounced. In 2D coculture TRAP activity decreased under hyperglycemia, when compared to hyperinsulinemia. In 3D coculture, similar to CAII activity, TRAP activity also showed an increasing trend under pre-/diabetic conditions (Figure 6C,D). In 2D coculture CTSK levels decreased under pre-/diabetic conditions, significantly under hyperglycemic conditions. Inversely to CAII and TRAP activity, in 3D coculture CTSK levels decreased under pre-/diabetic conditions (Figure 6E,F). As a consequence, collagen degradation, detected by NTX levels, was decreased under hyperglycemia in 2D coculture. In 3D coculture, NTX levels were barely affected by pre-/diabetic conditions (Figure 6G,H).

### 2.7. Pre-/diabetic Conditions Contrarily Affect Matrix Mineralization in 2D and 3D Coculture

In 2D coculture formation of mineralized matrix was significantly decreased under pre-/diabetic conditions, as determined by Alizarin Red staining (Figure 7A). However, in 3D coculture quantitative CT scans revealed a significant increase in the populated scaffolds’ mineral content under pre-/diabetic conditions. The effect was more pronounced under hyperglycemia than under hyperinsulinemia (Figure 7B). Additional measurement of the populated scaffolds’ stiffness revealed that pre-/diabetic conditions affected the stiffness negatively. Again, the effect was most pronounced under hyperglycemia (Figure 7C).

## 3. Discussion

In the present work, we aimed at developing an in vitro model that can display alterations in bone metabolism characteristic for type 2 diabetes mellitus. In previous studies, the effects of pre-/diabetic conditions have been investigated in mono-cultures with osteogenic cells [11,12,20,21]. In these models, a decrease in mineralized matrix formation was observed under disease conditions, which does not resemble the in vivo situation where type 2 diabetics are frequently associated with an increased BMD [4,5]. So far, no in vitro model could depict this apparent contradiction of increased bone mineral density and increased fracture risk characteristic for type 2 diabetics—challenging the used models [10]. As both precursor cells and more mature osteoblasts show decreased function, especially under diabetic conditions the presence of osteoclastic cells might be required. Furthermore, the stability of the formed matrix is hard to analyze in conventional 2D culture, therefore, a 3D carrier might be required.

Aiming for the possibility to broadly use the established in vitro bone metabolism model for screening purposes, we based the coculture on cell lines [14]. Formation of bone matrix is provided by osteoblastic SaOS-2 cells. As representatives of the osteoclast lineage, myeloid THP-1 cells were used. In coculture, SaOS-2 cells stimulate THP-1 cells with M-CSF and sRANKL to differentiate osteoclastogenesis [13,14,22]. The direct coculture of these two cells stabilized the system. While monocultures of the respective cell types can only be kept for 7–10 days [13,19], the coculture was stable for at least 2 weeks. Similar to the SaOS-2 cell monocultures [19], osteocalcin expression was high from the beginning of the culture. ALP activity rapidly decreased with increasing formation of mineralized matrix in the 2D coculture. This effect was delayed under pre-/diabetic conditions, suggesting towards a delay in osteogenic maturation. This is in line with earlier studies investigating the hyperglycemia and/or hyperinsulinemia on preosteoblasts, osteoblasts or osteocytes [11,12,20,21,23]. Overall, in these studies osteogenic function was reduced under pre-/diabetic conditions. While in less mature cells induction of proliferation seemed to delay the differentiation process [11,12,21], in mature osteocyte-like MLO-Y4 cells a disturbance in cell–cell gap-junctions was proposed as a possible mechanism for the disturbed function [20]. Interestingly, in 2D coculture presence of hyperglycemia and hyperinsulinemia resulted in increased numbers of THP-1 cells. This is in line with a study differentiating osteoclasts from blood cells of type 2 diabetics and db/db mice. This study suggests, that poor glycemic control may induce osteoclast formation [24]. However, a meta-analysis, summarizing the results of 66 studies revealed an overall reduced bone turnover with lowered collagen degradation in diabetic patients [25]. In our 2D coculture, the collagen degradation marker NTX was decreased under pre-/diabetic conditions, indicating towards a decreased osteoclast activity, despite the increased numbers of THP-1 cells. This might result from the decrease of sRANKL and simultaneous increase of its antagonist OPG observed under pre-/diabetic conditions. The observed decrease in sRANKL is in line with serum levels of type 2 diabetics, associated with a low bone turnover and reduced numbers of mature osteoblasts [26]. However, it is also in contrast to observations in MC3T3-E1 cells, where hyperglycemia was reported to induce expression of both *OPG* and *RANKL* [27], suggesting that hyperglycemia is not the only regulator in diabetic bone disease. In addition to an increase in OPG, antagonizing sRANKL, it has been reported that the direct effect of RANKL on preosteoclasts is altered under hyperglycemia, inhibiting osteoclast fusion [28]. However, there is evidence that osteoclast fusion strongly depends on the surface the cells attach to [29]. Only when both organic and inorganic bone matrix are present expression of *Annexin A8* increased, which is required for the initial cell fusion [29]. Although SaOS-2 cells are reported to quickly produce mineralized matrix [19], it is conceivable that plating the THP-1 cells first is not ideal in case of the 2D coculture. Considering that both organic and inorganic bone matrix are required for initial fusion of osteoclastic cells, this might contribute to the reduced osteoclast function observed in the 2D coculture. Under these circumstances both osteoblast function and osteoclast function were inhibited in the presence of hyperglycemia and hyperinsulinemia, overall resulting in decreased mineralized matrix. Being contrary to the in vivo situation, this raised the question whether using a suitable 3D matrix could be beneficial for the coculture.

Despite the advantages for osteoclast fusion, 3D carriers have been described to favor osteogenic differentiation of the applied cells. This might occur either directly by interaction of the cells with the carrier or indirectly by paracrine factors [15]. The here described model is based on pHEMA cryogels, which have been used as 3D carriers in a multitude of in vitro and in vivo experiments. Altering the pHEMA cryogels composition supported the osteogenic differentiation of applied adipose-derived MSCs in preliminary experiments [18]. For example, replacing insoluble hydroxyapatite by externally crystallized calcium phosphates increased the surface roughness of the carrier, a factor well described to favor cell attachment, proliferation, as well as osteogenic and osteoclastic differentiation [30]. By additionally crosslinking human platelet-rich plasma (PRP) to the scaffold, pore size and scaffold stiffness were further improved [18], reaching a mean stiffness close to the stiffness of collagenous bone [31,32]. Therefore, this scaffold represented an ideal carrier for osteogenic progenitor cells, osteoblastic cells and osteoclastic cells. In respective monocultures, the PRP scaffold proved to be advantageous compared to a more simple gelatin based cryogel, which could not show osteoclast-induced degradation of the bone matrix, despite a higher basal mineral content [17]. SaOS-2 cells cultured on the PRP scaffold showed both increased ALP activity and collagen production (PINP levels) [17], this might be partly due to the rough surface of the cryogels reported to induce ALP activity [33]. Therefore, it is not surprising that the decrease in ALP activity was delayed in 3D coculture when compared to 2D coculture. Considering the actual ALP activity, basal levels were higher in 2D coculture, which might be an effect of the larger reaction volume in 3D in combination with the stiff culture plastic [34]. In contrast to 2D coculture, 3D coculture showed a decrease in ALP activity in the presence of high glucose, indicating towards a decreased cryogel stiffness. In contrast OPG, expression of which is also directly associated with bone stiffness [35], increased under hyperinsulinemia. However, surface stiffness is not the only factor affecting OPG levels. OPG expression positively correlated with the porosity [34] and the surface roughness [36] of the 3D carrier. In 3D coculture, presence of insulin and glucose not only stabilized the amount of THP-1 cells but also strongly induced proliferation of SaOS-2 cells, which affects physical characteristics of the scaffolds [17]. In contrast to 2D coculture, sRANKL levels increased under hyperglycemia and 3D conditions, supporting osteoclast formation, which would be in clear contrast to the in vivo situation [26]. However, CAII and TRAP activity show only a mild trend to increase under pre-/diabetic conditions in 3D coculture, only CTSK levels increased under these conditions. It has been shown that hyperglycemia may affect sRANKL induced osteoclast formation by interfering with NF-kB signaling [37]. However, the decrease in CAII and TRAP activity observed in 2D coculture is missing in 3D coculture, underlining the need for both the organic and the inorganic bone matrix to induce cell fusion of osteoclasts [29]. Altering the balance between the organic matrix and the soluble calcium available is thought to regulate osteoclast activity and apoptosis [38]. This might be triggered by proliferating SaOS-2 cells in the 3D coculture under prediabetic conditions. Strong proliferation is often associated with delayed maturation, which might explain the large amount of osteogenic precursor cells observed in type 2 diabetics [26]. Overall, the observed changes in osteoblastic and osteoclastic function result in increased formation of mineralized matrix under pre-/diabetic conditions in 3D coculture. Accumulation of mineralized matrix might induce expression of sclerostin, described to be increased in type 2 diabetics [39], by osteocytes, thus, actively inhibiting maturation of the osteogenic precursor cells by inhibiting Wnt signaling. The presence of the 3D carrier allows measuring alterations of the materials’ stiffness [17,18]. Despite the increase in mineralized matrix observed under pre-/diabetic conditions the stability of the 3D carrier decreased, resembling the clinical situation in type 2 diabetics [4,5]. In summary, our results clearly show that the 3D environment is required to display the anticipated changes in bone metabolism in this in vitro model, providing the option to more generally use the described in vitro model, e.g., for investigating alteration in bone metabolism in other metabolic bone diseases, for screening substances or drugs, or for testing treatment option [10]. A factor not addressed by this in vitro model is the increased bone marrow adiposity, frequently observed in type 2 diabetics [39]. Addressing this would require the addition of, or even the replacement of the mature SaOS-2 cells by more immature osteoprogenitor cells, e.g., the SCP-1 cells [40], which proved to be well compatible with the here used THP-1 cells [14].

## 4. Materials and Methods

Chemicals and reagents were obtained from Carl Roth (Karlsruhe, GER) or Sigma-Aldrich/Merck (Darmstadt, GER). Culture medium and its supplements were obtained from Sigma-Aldrich/Merck.

### 4.1. Scaffold Manufacturing and Sterilization

Cryogel-scaffolds were prepared as previously described [17,18]. Briefly, an aqueous solution containing 16.0% pHEMA, 0.3% N,N-methylene(bis)acrylamide (BAAm), and 0.25 g/L platelet-rich plasma (PRP) was carefully mixed and cooled on ice. After 30 min, di-sodium hydrogen phosphate buffer was added to obtain a final solution of 0.3 M. Immediately after adding 0.1% glutaraldehyde, 0.2% ammonium persulfate (APS), and 0.2% N,N,N,N-tetra methyl-ethylene diamine (TEMED), the reaction solution was mixed, distributed into polystyrene casting molds (6 mm inner diameter/2 mL per mold) and frozen at −18 °C for at least 17 h. The polymerized matrix was deep-frozen for 1 h at −80 °C in order to ease slicing with a razor blade (3 mm height). The frozen pHEMA-based scaffolds were immediately transferred to a 1 M CaCl_2_ solution to facilitate crystallization of calcium phosphate. After 24 h CaCl_2_ solution was carefully aspirated and the scaffolds were immersed in 70% ethanol for at least 12 h for sterilization. After three washing steps with PBS (1, 6, and 12 h), sterilized scaffolds were incubated in culture medium for 48 h (37 °C, 5% CO_2_, humidified atmosphere) for preconditioning and as sterility control. Pore size, porosity, water uptake rate, and permeability of the generated scaffolds have been described in detail before [17,18].

### 4.2. Cell Lines

THP-1 cells (DSMZ) were used as osteoclastic precursor cells and SaOS-2 cells (DSMZ) were used representative for osteogenic cells [14]. Both cell lines were cultured in RPMI 1640 Medium with 5% FCS (37 °C, 5% CO_2_, humidified atmosphere). Medium was changed twice a week.

### 4.3. 2D Coculture

The 2 × 10^4^ THP-1 cells were applied to each cavity of a 96-well plate. Medium contained 200 nM PMA to adherence of the cells. After 24 h at 37 °C (5% CO_2_, humidified atmosphere), medium was thoroughly aspirated and 1 × 10^4^ SaOS-2 cells in 100 µL osteogenic medium (RPMI 1640, 2% FCS, 200 μM L-ascorbic acid 2-phosphate, 5 mM β-glycerol phosphate, 25 mM HEPES, 1.5 mM CaCl_2_, and 5 μM cholecalciferol [17]) were applied to each cavity of the 96-well plate. 2D cocultures were maintained at 37 °C (5% CO_2_, humidified atmosphere). Osteogenic medium was replaced on days 1, 4, 7, and 11 of culture.

### 4.4. 3D Coculture

To obtain the required coculture in 3D, cells were also sequentially seeded on the scaffolds. Medium was thoroughly aspirated from the preconditioned scaffolds. One scaffold each was placed centrally in the cavities of a 48-well plate. A total of 15 µL of a THP-1 cell-solution (8 × 10^6^ cells/mL) containing 200 nM PMA was dripped centrally on top of each scaffold to obtain a seeding density of 1.2 × 10^5^ cells/scaffold. After an initial incubation of 4 h at 37 °C (5% CO_2_, humidified atmosphere), 505 μL of the respective cell culture medium was carefully added. After 24 h (37 °C, 5% CO_2_, humidified atmosphere), medium was again carefully aspirated, and scaffolds were washed once with PBS to ensure removal of the PMA. Then, 15 µL of a SaOS-2 cell-solution (4 × 10^6^ cells/mL) was dripped centrally on top of each scaffold to obtain a seeding density of 6 × 10^4^ cells/scaffold. After another 4 h incubation at 37 °C (5% CO_2_, humidified atmosphere), 505 μL of osteogenic medium (RPMI 1640, 2% FCS, 200 μM L-ascorbic acid 2-phosphate, 5 mM β-glycerol phosphate, 25 mM HEPES, 1.5 mM CaCl_2_, and 5 μM cholecalciferol) was added [17]. Maintenance and medium changes of 3D cocultures were comparable to the 2D cocultures.

### 4.5. In vitro Conditions to Simulate the Development of a Type 2 Diabetes Mellitus

To simulate the development of a type 2 diabetes mellitus in vitro, glucose and insulin were supplemented to the medium as described before [11]:
For normoglycemic control conditions (CO), 14 mM mannitol was added to the osteogenic medium (11 mM glucose) to equalize osmolality compared to hyperglycemic conditions.For the “prediabetic conditions” with high insulin (HI), CO medium was supplemented with 160 I.U./l insulin (Actrapid, NovoNordisk, Bagsværd, DNK).For the “diabetic conditions” with hyperglycemia (HG), medium was additionally supplemented with 14 mM glucose, to obtain a final glucose concentration of 25 mM.

### 4.6. Resazurin Conversion Assay

Mitochondrial activity of the cells was determined by Resazurin conversion assay. Cells in 2D and 3D were washed once with PBS. Populated scaffolds were transferred to a new 48-well plate to prevent false-positive signals from cells attached to the culture plastic. Resazurin solution (0.0025% in plain RPMI 1640 Medium, 2D = 100 µL and 3D = 520 µL) was added to the cells and blank (empty well ± unpopulated scaffold). After incubation for 2 h at 37 °C the produced Resorufin was quantified by its fluorescence at λ_ex_ = 544 nm and λ_em_ = 590–10 nm using the Omega Plate Reader (BMG Labtech, Ortenberg, GER). To do so for 3D cocultures, 2 × 100 µL of each sample were transferred into cavities of 96-well-plate [17,18].

### 4.7. Isolation of Total DNA

To isolate the total DNA, samples were washed once with PBS. Liquid was thoroughly removed from the scaffolds by centrifugation (600× *g*, 10 min) with a cell strainer. Scaffolds without cells were used as background control. Samples were incubated in 50 mM hot (98 °C) NaOH (50 µL for 2D and 259 µL for 3D) for 15 min. Then, samples were frozen at –80 °C for at least 24 h. After thawing at 60 °C for 20 min, an equal volume of a 100 mM Tris buffer (pH 8.0) was added to each sample to neutralize the pH. These samples were centrifuged at 1.000× *g* for 10 min to remove impurities [18].

### 4.8. Quantification of Cell-Specific DNA

Quantification of cell-specific DNA was performed by SYBR-green based qPCR using sex-specific primers [10]. Total DNA was detected using primers for *UGT1A6* (uridine diphosphate glucuronosyl transferase 1A6), located on chromosome 2: forward-TGGTGCCTGAAGTTAATTTGCT and reverse-GCTCTGGCAGTTGATGAAGTA; 60 °C annealing temperature; 209 bp amplicon size [41]. The amount of male DNA (THP-1 cells) was detected using primers for *SRY* (sex-determining region Y), located on the Y-chromosome: forward-TGGCGATTAAGTCAAATTCGC and reverse-CCCCCTAGTACCCTGA CAATGTATT; 60 °C annealing temperature; 137 bp amplicon size [42]. Standard curves using the individual cell lines were used for normalization.

### 4.9. Enzyme-Linked Immunosorbent Assay (ELISA)

Target proteins in the undiluted culture supernatants were quantified with the help of ELISA kits, performed as indicated by the manufacturer. For overview see Table 1.

### 4.10. Dot Blot Analysis

Specific proteins in the culture supernatants were detected by dot blot. Briefly, 70 µL of the cell culture supernatant was applied on a wet nitrocellulose membrane with the help of a dot blotter (Carl Roth). After blocking membranes with 5% BSA for 1 h, primary antibody incubation (Cathepsin K—CTSK—sc-48353 and osteocalcin—OC—sc-365797; both obtained from Santa Cruz Biotechnology, Heidelberg, GER) was performed at 4 °C overnight. After incubation with the corresponding peroxidase-labeled secondary antibodies (Santa Cruz Biotechnology, Heidelberg, GER) for 2 h, chemiluminescent signals were detected by a CCD camera (INTAS, Göttingen, GER) and quantified using the ImageJ software [17,18].

### 4.11. Alkaline Phosphatase (ALP) Activity

ALP activity was determined by conversion of p-nitrophenyl phosphate (pNPP) into p-nitrophenol (pNP). After removal of the medium, cells (± scaffold) were incubated with substrate solution (1 mg/mL pNPP, 50 mM glycine, 1 mM MgCl_2_, 100 mM TRIS; pH 10.5; 2D: 100 µL vs. 3D: 520 µL). Formed pNP was detected photometrically at λ = 405 nm using the Omega Plate Reader: in 2D cultures as a kinetic over 15 min at 37 °C, and in 3D cultures by transfer of 100 µL of the reaction solution to a fresh 96-well plate with endpoint measurement after 5 min of incubation at 37 °C [18].

### 4.12. Tartrate-Resistant Acid Phosphatase (TRAP) Activity

TRAP activity was detected in the culture supernatant by conversion of pNPP into pNP in an acidic environment and in presence of tartrate. Briefly, 30 µL culture supernatant were mixed with 90 µL reaction solution (100 mM sodium acetate, 50 mM di-sodium tartrate, 5 mM pNPP, pH 5.5). After 6 h incubation at 37 °C the reaction was stopped by adding 90 µL 1 M NaOH. Formed pNP was detected photometrically at λ = 405 nm using the Omega Plate Reader.

### 4.13. Carbonic Anhydrase II (CAII) Activity

CAII activity was determined by conversion of 4-nitrophenyl acetate (pNPA) into pNP by the cells [43]. Briefly, after thorough removal of the medium, cells (± scaffold) were incubated with substrate solution (2 mM pNPA, 75 mM NaCl, 12.5 mM TRIS; pH 7.5; 2D: 100 µL vs. 3D: 520 µL). Photometric detection of the formed pNP at λ = 405 nm was performed using the Omega Plate Reader: similar to ALP activity, a kinetic over 15 min at 37 °C was measured for 2D cultures. For 3D cultures 100 µL of the reaction solution was transferred to a fresh 96-well plate for endpoint measurement after 15 min of incubation at 37 °C.

### 4.14. Formation of Mineralized Matrix

#### 4.14.1. Alizarin Red Staining in 2D Cocultures

Formation of mineralized matrix in 2D cultured was determined by Alizarin Red staining. Briefly, cells were fixed for 1 h with ice-cold 99.9% ethanol (–20 °C). After washing three times with tap water, cells were incubated with 0.5% Alizarin Red solution (pH 4.0) for 30 min at room temperature. After three additional washing steps, the resulting staining was assessed microscopically. Bound Alizarin Red was resolved with a 10% Cetylpyridium chloride solution for photometric quantification at λ = 562 nm using the Omega Plate Reader. Formation of mineralized matrix in 3D cocultures could not be assessed by this technique, as the stain would bind to the mineralized cryogels.

#### 4.14.2. Mineral Content of the Scaffolds (3D Cocultures)

The mineral content of the scaffolds after 2 weeks of culture was determined using a high-end 128-slice clinical computed tomography (CT) scanner (Somatom Definition Edge, Siemens Healthineers, Erlangen, GER). Image acquisition was performed using a tube voltage of 80 kV, a fixed effective tube current of 500 mAs. Primary acquisition was performed with 16 × 0.3 mm and a pitch of 0.4 and rotation time of 1.0 s, reconstructed slice thickness was 0.4 mm. Images were reconstructed with a squared field-of-view of 50 mm at a resolution of 512 × 512 pixels. Image reconstruction as performed using a sinogram affirmed iterative reconstruction (SAFIRE, Siemens Healthineers, Erlangen, GER) at level 5 with a sharp edge enhancing reconstruction kernel (v80U); images were displayed in bone window. Obtained DICOM images were imported into the ImageJ software using the “DICOM sort” plugin. The resulting stack was cropped to show the area of interest. From each scaffold, the mean grey values were determined and normalized to values of the reference block (Phantom EFP-06-96) [17].

### 4.15. Scaffold Stiffness

The scaffold stiffness was determined by the Young’s modulus, measured using a ZwickiLine Z 2.5TN (Zwick GmbH & Co.KG, Ulm, GER) material testing machine [17]. Briefly, scaffolds were compressed four times uniaxial by 10% (5 mm/min speed) of the original height. A Xforce HP 5N sensor measured the required load in real-time. The resulting load-deformation curve was translated into a stress–strain curve by using the height and area of the uncompressed scaffold. The Young’s modulus (MPa) in the region of linear elastic deformation was calculated by dividing the applied force (N) times the initial scaffold height (mm) by the area of the scaffold (mm^2^) times the change in height (mm).

### 4.16. Statistical Analysis

Results are presented as bar diagrams or curves (mean ± SEM). Each experiment was performed three or four times (N = 3 or 4), with at least two technical replicates (n ≥ 2). The number of biological and technical replicates is given in the figure legends. Statistical analyses were performed using the GraphPad Prism Software version 8 (GraphPad, El Camino Real, USA). Data were compared by nonparametric Friedmann test for paired data, followed by Dunn’s multiple comparison test. Changes in cell-specific DNA were analyzed by nonparametric two-way ANOVA, followed by Tukey’s multiple comparison test. A *p*-value below 0.05 was considered statistically significant.

## 5. Conclusions

In the here described in vitro model, formation of bone matrix is provided by osteoblastic SaOS-2 cells. In coculture, these cells stimulate THP-1 cells to differentiate to osteoclastic cells, able to degrade bone matrix. Our results clearly show that the coculture is more stable than the respective monocultures of these cells. Furthermore, the 3D environment is required to display the alterations in bone metabolism characteristic for type 2 diabetes mellitus. While in 2D coculture formation of mineralized matrix is decreased under pre-/diabetic conditions, formation of mineralized matrix is increased in 3D coculture. Furthermore, despite the increase in mineralized matrix stability of the 3D carrier decreased under pre-/diabetic conditions, resembling the clinical situation in type 2 diabetics. Based on the here presented data, a more general use of the here described 3D coculture model, for example investigating effects of other diseases, treatments, substances, or drugs on bone metabolism in vitro, is feasible.

## Figures and Tables

**Figure 1 ijms-22-02925-f001:**
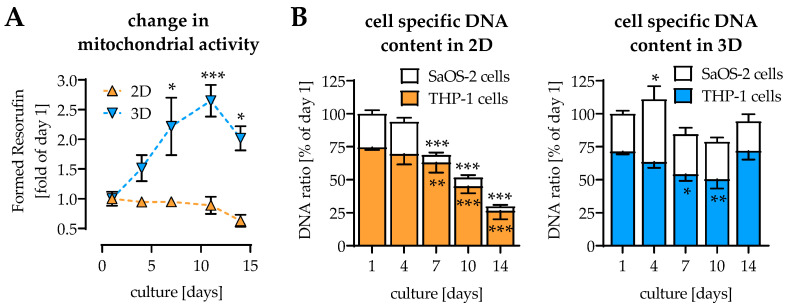
Comparison of viability of THP-1 cells and SaOS-2 cells in 2D and 3D coculture. THP-1 cells and SaOS-2 cells (2:1 ratio) were cocultured over 14 days in 2D and 3D on pHEMA-based cryogel scaffolds. On days 1, 4, 7, 10, and 14 of culture, (**A**) mitochondrial activity (Resazurin conversion) and (**B**) cell-specific DNA content were measured. Data are presented as curves or bar charts (mean ± SEM). Changes over time were compared by nonparametric Friedmann test, followed by Dunn’s multiple comparison test. Cell-specific DNA contents were compared by nonparametric two-way ANOVA, followed by Tukey’s multiple comparison test. * *p* < 0.05, ** *p* < 0.01, and *** *p* < 0.0001 as compared to the respective day 1. *N* ≥ 3 and *n* = 3.

**Figure 2 ijms-22-02925-f002:**
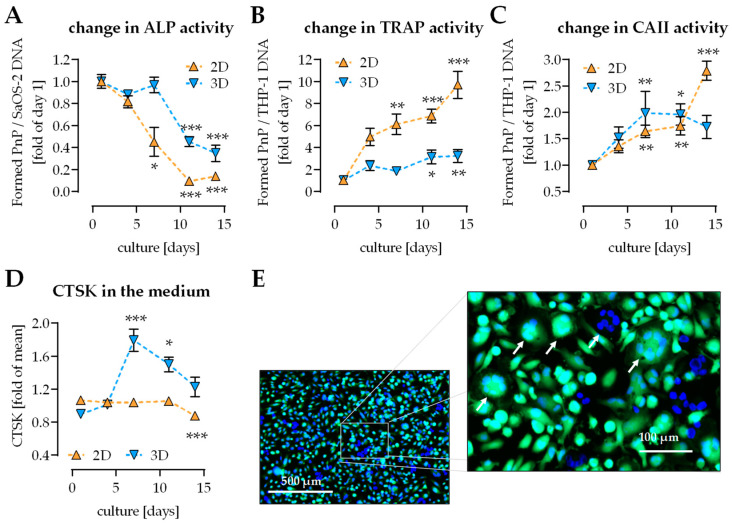
Comparison of functionality of THP-1 cells and SaOS-2 cells in 2D and 3D coculture. THP-1 cells and SaOS-2 cells (2:1 ratio) were cocultured over 14 days in 2D and 3D on pHEMA-based cryogel scaffolds. On days 1, 4, 7, 10, and 14 of culture, (**A**) alkaline phosphatase (ALP) activity, (**B**) tartrate-resistant acidic phosphatase (TRAP) activity, (**C**) carbonic anhydrase II (CAII) activity, and (**D**) cathepsin K (CTSK) levels were measured. (**E**) On day 7 life-dead-staining revealed the presence of large multinucleated cells (arrow). Data are presented as curves (mean ± SEM). Changes over time were compared by nonparametric Friedmann test, followed by Dunn’s multiple comparison test. * *p* < 0.05, ** *p* < 0.01, and *** *p* < 0.0001 as compared to the respective day 1. *N* ≥ 3 and *n* = 3.

**Figure 3 ijms-22-02925-f003:**
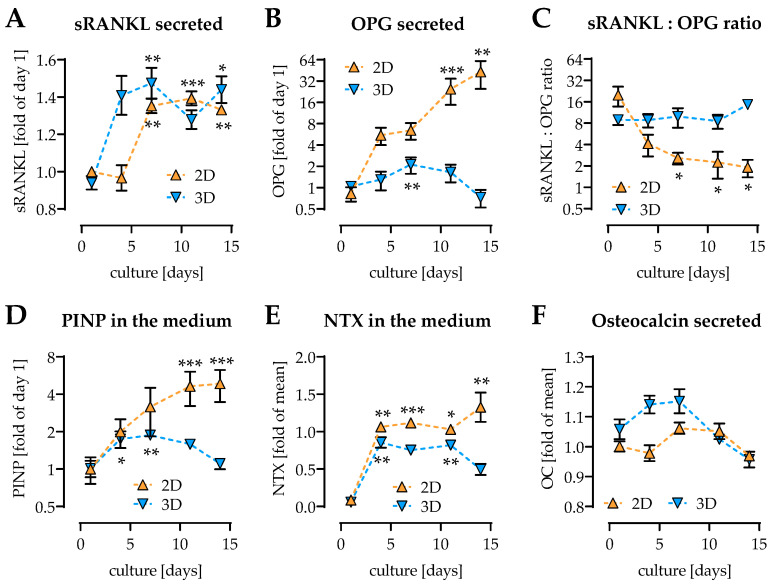
Secreted markers for osteoblast and osteoclast function in 2D and 3D cocultures of THP-1 cells and SaOS-2 cells. THP-1 cells and SaOS-2 cells (2:1 ratio) were cocultured over 14 days in both 2D and 3D on pHEMA-based cryogel scaffolds. On days 1, 4, 7, 10, and 14 of culture, (**A**) soluble receptor activator of nuclear factor kappa-Β ligand (sRANKL), (**B**) osteoprotegerin (OPG), and (**C**) the sRANKL:OPG ratio, (**D**) procollagen type I N-terminal propeptide (PINP), (**E**) collagen-type I N-telopeptide (NTX), and (**F**) osteocalcin levels were detected in the culture supernatant. Data are presented as curves (mean ± SEM). Changes over time were compared by nonparametric Friedmann test, followed by Dunn’s multiple comparison test. * *p* < 0.05, ** *p* < 0.01, and *** *p* < 0.0001 as compared to day 1. *N* = 3, *n* = 2.

**Figure 4 ijms-22-02925-f004:**
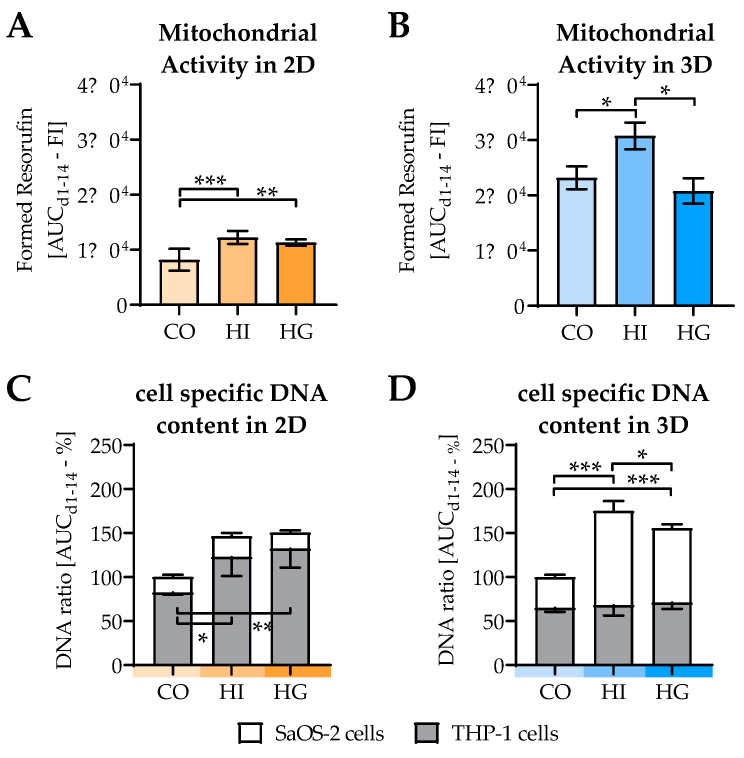
Influence of pre-/diabetic conditions on viability of THP-1 cells and SaOS-2 cells in 2D and 3D cocultures. THP-1 cells and SaOS-2 cells (2:1 ratio) were cocultured over 14 days in both (**A**,**C**) 2D and (**B**,**D**) 3D on pHEMA-based cryogel scaffolds in the presence or absence of high insulin levels (HI = prediabetic conditions) or high glucose levels (HG = diabetic conditions). On days 1, 4, 7, 10, and 14 of culture, (**A**,**B**) mitochondrial activity (Resazurin conversion) and (**C**,**D**) cell-specific DNA content were determined. To consider the entire differentiation period, data are presented as area under the curve (AUC/mean ± SEM). Mitochondrial activities were compared by nonparametric Friedmann test, followed by Dunn’s multiple comparison test. Cell-specific DNA contents were compared by nonparametric two-way ANOVA, followed by Tukey’s multiple comparison test. * *p* < 0.05, ** *p* < 0.01, and *** *p* < 0.0001 as indicated. *N* = 3, *n* = 3.

**Figure 5 ijms-22-02925-f005:**
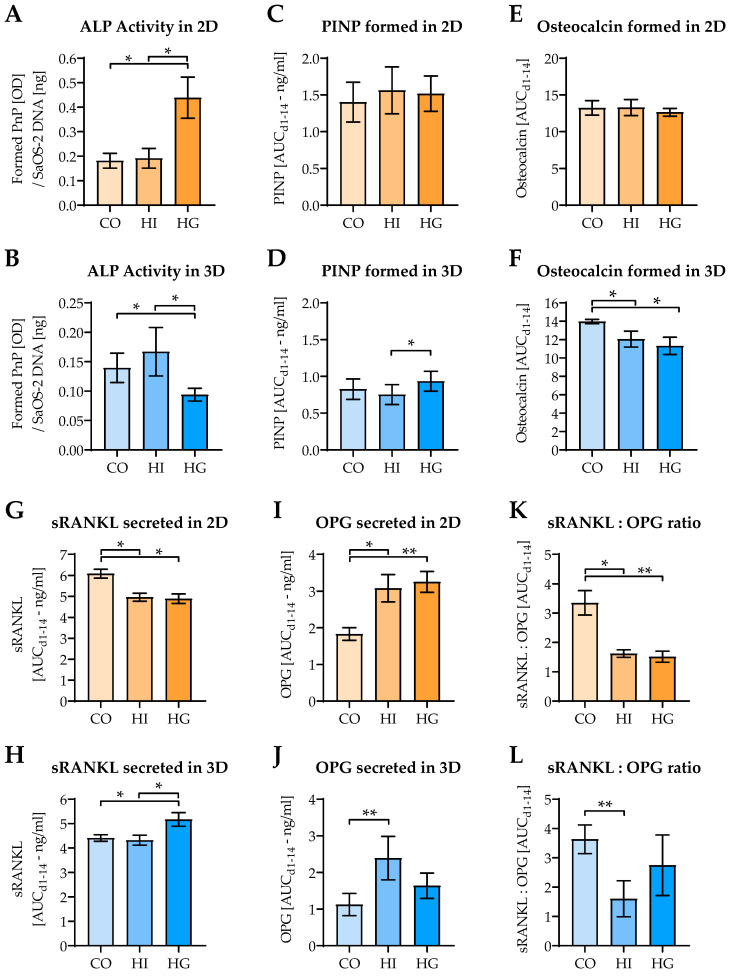
Influence of pre-/diabetic conditions on SaOS-2 cells’ function in 2D and 3D cocultures. THP-1 cells and SaOS-2 cells (2:1 ratio) were differentiated for 14 days as (**A**,**C**,**E**,**G**,**I**,**K**) 2D and (**B**,**D**,**F**,**H**,**J**,**L**) 3D cocultures in the presence or absence of high insulin levels (HI = prediabetic conditions) or high glucose levels (HG = diabetic conditions). On days 1, 4, 7, 10, and 14 of culture, (**A**,**B**) alkaline phosphatase (ALP) activity was measured. At the same time points (**C**,**D**) procollagen type I N-terminal propeptide (PINP), (**E**,**F**) osteocalcin, (**G**,**H**) soluble receptor activator of nuclear factor kappa-Β ligand (sRANKL), (**I**,**J**) osteoprotegerin (OPG) levels, and (**K**,**L**) the sRANKL:OPG ratio were detected in the culture supernatants. ALP activity was determined on the day of culture before it dropped significantly. For the other factors the entire differentiation period was considered, and data are presented as area under the curve (AUC/mean ± SEM). Data were compared by nonparametric Friedmann test, followed by Dunn’s multiple comparison test. * *p* < 0.05 and ** *p* < 0.01 as indicated. *N* = 3, *n* = 3.

**Figure 6 ijms-22-02925-f006:**
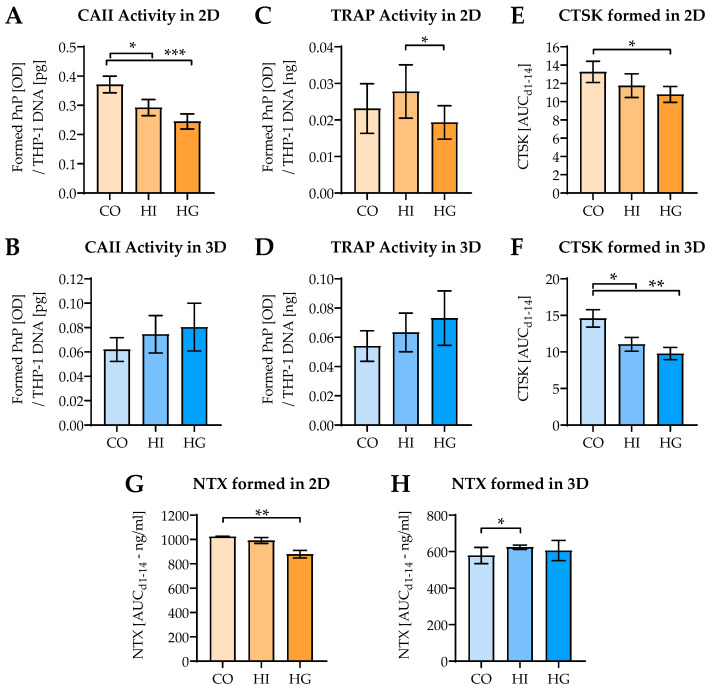
Influence of pre-/diabetic conditions on THP-1 cells’ function in 2D and 3D cocultures. THP-1 cells and SaOS-2 cells (2:1 ratio) were cocultured for 14 days in (**A**,**C**,**E**,**G**) 2D and (**B**,**D**,**F**,**H**) 3D, in the presence or absence of high insulin levels (HI = prediabetic conditions) or high glucose levels (HG = diabetic conditions). On days 1, 4, 7, 10, and 14 of culture (**A**,**B**) carbonic anhydrase II (CAII) activity and (**C**,**D**) tartrate-resistant acidic phosphatase (TRAP) activity were measured. At the same time points (**E**,**F**) cathepsin K (CTSK) and (**G**,**H**) collagen-type I N-telopeptide (NTX) levels were detected in the culture supernatants. CAII and TRAP activities are displayed on their day of maximal activity. CTSK and NTX levels are presented as area under the curve (AUC/mean ± SEM), to consider all measured time points. Data were compared by nonparametric Friedmann test, followed by Dunn’s multiple comparison test. * *p* < 0.05, ** *p* < 0.01, and *** *p* < 0.0001 as indicated. *N* = 3, *n* = 3.

**Figure 7 ijms-22-02925-f007:**
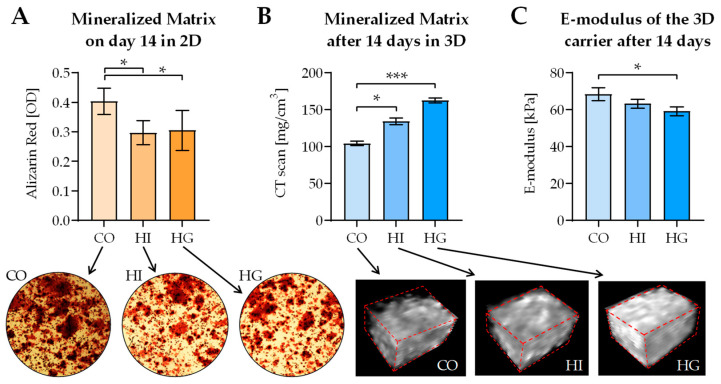
Influence of pre-/diabetic conditions on the formation of mineralized matrix and matrix stiffness in 2D and 3D cocultures of THP-1 cells and SaOS-2 cells. THP-1 cells and SaOS-2 cells (2:1 ratio) were differentiated for 14 days as (**A**) 2D and (**B**,**C**) 3D coculture, in the presence or absence of high insulin levels (HI = prediabetic conditions) or high glucose levels (HG = diabetic conditions). After 14 days of culture matrix mineralization was quantified by (**A**) Alizarin Red staining in 2D with representative microscopic image and (**B**) quantitative computer tomography (CT) scans in 3D, with a representative 3D reconstruction of the central core of the scaffold. (**C**) In 3D cocultures matrix stiffness (E-modulus) was also determined using a ZwickiLine Z 2.5TN material testing machine. Data, represented as mean ± SEM, were compared by nonparametric Friedmann test, followed by Dunn’s multiple comparison test. * *p* < 0.05 and *** *p* < 0.0001 as indicated. *N* = 3, *n* = 3.

**Table 1 ijms-22-02925-t001:** Overview on the used ELISA kits.

Target Protein	Role	Order #	Company
NTX (collagen-type I N-telopeptide)	Collagen degradation	E-EL-H0836	Elabscience, Houston, TX, USA
OPG (osteoprotegerin)	Inhibitor for sRANKL	ABIN411341	Antibodies-online, Aachen, Germany
PINP (procollagen type I N-terminal propeptide)	Collagen formation	8003	TecoMedical, Neufahrn, Germany
sRANKL (soluble receptor activator of nuclear factor kappa-Β ligand	Inducer for osteoclastogenesis	900-K142	Peprotech, Hamburg, Germany

## Data Availability

The datasets generated during and/or analyzed during the current study are available from the corresponding author on reasonable request.

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
