# Peer review of "3D Environment Is Required In Vitro to Demonstrate Altered Bone Metabolism Characteristic for Type 2 Diabetics"

_ijms, 2021, doi:10.3390/ijms22062925_

Round 1

Reviewer 1 Report

In the present work the authors  described an in vitro model displaying bone metabolism frequently observed in diabetics. The model is based on osteoblastic SaOS-2 cells, which in direct co-culture, stimulate THP-1  cells to form osteoclasts. This study aimed at establishing an in vitro model displaying changes in  bone metabolism characteristic for type 2 diabetics, which involves both bone formation  by osteogenic cells and bone resorption by osteoclastic cells.

In abstract section

It is strongly suggested to revise the abstract section and to improve the quality of english.  The sentences are not clear. 

In the introduction section:

line 114-115: Aiming to develop an in vitro model, which can display changes in bone metabolism characteristic for type 2 diabetics, pre-/diabetic conditions will be simulated as described, in both the proposed  2D and 3D co-cultures., please this sentence need to be revised in order to be clearer.

Figure 4, C: please revise the figure C, the significativity. 

Overall the study is well performed and the results obtained are interesting.

Major concerns:

it is suggested to the Author to add some pictures made at optical microscope or confocal to show the morphological changes at the different conditions and time points.

For example, the authors have validate the mineralization through Alizarid red, it is strongly suggest to add some pictures of the Alizarin red staining. 

the manuscript need to be revised by a native english speaker. 

Author Response

Please find the detailed answers to the reviewer's comments in the attached PDF.

Reviewer 2 Report

The study described an in vitro model displaying bone metabolism frequently observed in diabetics. The model was established on osteoblastic SaOS-2 cells, which in direct co-culture, stimulated THP-1 cells to form osteoclasts. The authors found that in conventional 2D co-cultures formation of mineralized matrix was decreased under pre-/diabetic conditions, while formation of mineralized matrix was increased in 3D cocultures. Furthermore, they demonstrated a matrix stability of the 3D carrier decreased under pre-/diabetic conditions, resembling the in vivo situation in type 2 diabetics. They concluded that a 3D environment is required in this in vitro model to mimic alterations in bone metabolism characteristic for pre-/diabetes. The ability to measure both osteoblast and osteoclast function, and their effect on mineralization and stability of the 3D carrier offers the possibility to use this model also for other purposes, e.g. drug screenings. In general, this manuscript is well written and provides a structural and contextual understanding for the results, as well as the theories. The study may deserve publication after minor revision.

some comments and suggestions:

  1. There were comparisons between 2D and 3D groups, between different cells, and between different contents (Figure 1~7). What I cannot find in the study is the number of samples in each group to be compared to each a statistical significance. Otherwise the conclusion should be made more conservatively or at least, this condition should be stated in the section of the limitation of the research.
  2. Line 19: Abstract suggest rephrasing " Although there is steady progress in treating diabetes, the mechanisms why diabetes leads to massive bone alterations remain unknown.
  3. Line 21: Abstract “We herein...”

Author Response

Please find the detailed answers to the reviewer's questions in the attached PDF.

Reviewer 3 Report

The Article “3D environment is required in vitro to demonstrate altered bone metabolism characteristic for type 2 diabetics” describes new in vitro model that can monitor alterations in bone metabolism characteristic for type 2 DM.  Authors also mentions the possibility to use this model for drug screenings for diabetic bone disorder.  Findings in this study offer novel insights into the research field of bone metabolism.  This manuscript contains sufficient interest and originality to merit publication.  However, I have several concerns with this manuscript.  Please confirm my comments as indicated below.

  1. Introduction section could be substantially reduced and made more simple. It is quite long.

  1. Figure 2, 3, 5, 6: Did you check other osteoblastic and osteoclastic marker? Especially, I have interest about OCN and CtsK gene.  Can you show the expression analysis data of these genes by RT-PCR?

  1. Figure 3, 5: Authors should show the RANKL/OPG ratio.

  1. Figure 2, 6: Are there TRAP positive multinucleated cells? Can you show the morphology and number of TRAP-positive osteoclastic cells in cultured THP-1 cells.

  1. In paragraph 2.2, authors should clearly mention that what the ALP, TRAP and CAII activity means in Ob and Oc culture system.

Author Response

(The authors gave the same response as above.)

Round 2

Reviewer 1 Report

The authors have replied to all the comments, and they modified the manuscript accordingly. 

It is strongly suggest to improve the quality of the figures , the resolution is law, please improve the resolution of the figures. 

Moreover, the authors should provide a  better figure of the alizarin red staining, the rappresentative figures are too small.  

Author Response

As suggested by the reviewer, the figures were enlarged and embedded into the text with a higher resolution.

Reviewer 3 Report

The revised manuscript was improved. I recommend the accepting of the manuscript.

Author Response

We'd like to thank the reviewer for his/her recommendation.

As suggested by another reviewer, the figured were enlarged and embedded into the text with a higher resolution.